# Improving Specific Absorption Rate Efficiency and Coil Robustness of Self-Decoupled Transmit/Receive Coils by Elevating Feed and Mode Conductors

**DOI:** 10.3390/s23041800

**Published:** 2023-02-06

**Authors:** Ming Lu, Xiaoyang Zhang, Shuyang Chai, Xinqiang Yan

**Affiliations:** 1College of Nuclear Equipment and Nuclear Engineering, Yantai University, Yantai 264005, China; 2Vanderbilt University Institute of Imaging Science, Vanderbilt University Medical Center, Nashville, TN 37232, USA; 3Department of Radiology and Radiological Sciences, Vanderbilt University Medical Center, Nashville, TN 37232, USA; 4Department of Electrical and Computer Engineering, Vanderbilt University, Nashville, TN 37232, USA

**Keywords:** self-decoupled, SAR, RF coil, array, decoupling, high field

## Abstract

Self-decoupling technology was recently proposed for radio frequency (RF) coil array designs. Here, we propose a novel geometry to reduce the peak local specific absorption rate (SAR) and improve the robustness of the self-decoupled coil. We first demonstrate that B_1_ is determined by the arm conductors, while the maximum E-field and local SAR are determined by the feed conductor in a self-decoupled coil. Then, we investigate how the B_1_, E-field, local SAR, SAR efficiency, and coil robustness change with respect to different lift-off distances for feed and mode conductors. Next, the simulation of self-decoupled coils with optimal lift-off distances on a realistic human body is performed. Finally, self-decoupled coils with optimal lift-off distances are fabricated and tested on the workbench and MRI experiments. The peak 10 g-averaged SAR of the self-decoupled coil on the human body can be reduced by 34% by elevating the feed conductor. Less coil mismatching and less resonant frequency shift with respect to loadings were observed by elevating the mode conductor. Both the simulation and experimental results show that the coils with elevated conductors can preserve the high interelement isolation, B_1_^+^ efficiency, and SNR of the original self-decoupled coils.

## 1. Introduction

Magnetic resonance imaging (MRI) is a noninvasive medical imaging modality that can provide a variety of high-resolution and high soft-tissue contrast images [1]. Radiofrequency (RF) coils used in the MRI scanner play a critical role in determining image quality in terms of signal-to-noise ratio (SNR), image uniformity, etc. [2]. Transmit (Tx) RF coils are responsible for delivering the RF energy and thus exciting the spins in the sample, while receive (Rx) coils are responsible for detecting the MR signal from precessing magnetization. Array design is highly desired for RF Rx coils and high-field Tx coils. For Rx coils, the array design provides a high signal-to-noise ratio (SNR), flexible volume coverage, and encoding capability for fast imaging [3,4,5,6,7,8,9,10,11]. For Tx coils, the array design provides more freedom to manipulate the transmit field and specific absorption ratio (SAR) [12,13,14,15,16,17,18,19,20,21,22,23,24,25].

Decoupling is crucial to RF arrays because interelement coupling decreases the SNR and Tx efficiency, reduces the encoding capability, and makes individual B_1_ profiles less distinct. To date, many decoupling approaches have been proposed and used for coil arrays, such as geometric overlap, transformers, interconnecting L/C networks, and induced current elimination [3,26,27,28,29,30,31,32]. We recently proposed self-decoupled coils, which proved to be a simple and efficient approach to maintain extremely low interelement coupling without the need for any decoupling approaches [33]. In particular, they can be applied for Tx coils as well as Rx coils, as the mode of operation is independent of the subsequent circuit parameters, such as preamplifier impedance.

The self-decoupled coil uses intentionally uneven capacitor/current distributions along the conductor to generate dipole-mode (or electric) coupling to cancel the loop-mode (magnetic) coupling [33]. Our previous results revealed that it exhibits almost the same performance compared to an ideal conventional coil in terms of SNR, B_1_^+^ efficiency, and SAR efficiency when positioned several centimeters away from the loading [33]. The SAR efficiency is evaluated as the B_1_^+^ strength per root of the square of the maximum 10 g-averaged local SAR (maxSAR_10g_), representing the achievable B_1_^+^ for a given local SAR limit. Note that the SAR efficiency is also known as the B_1_^+^ SAR efficiency, and the two terms are interchangeably used here. When the self-decoupled coil was placed close to the loading, e.g., ~1 cm away from the loading in the transmit/receive applications, we noted that the strong current on the conductor near the feed port (herein referred to as the feed conductor) leads to a higher maximum local SAR. Meanwhile, we noted that the coil impedance and resonant frequency of self-decoupled coils are more sensitive to loading, partly because small mode capacitors (C_mode_) are more likely to be affected by the parasitic capacitance between the coil and loading.

When looking into the electromagnetic fields generated by different conductors in a self-decoupled coil, we found that (1) the rotating magnetic fields (B_1_ = B_x_ ± iB_y_) [34] are determined by the currents along the arm conductors; (2) the maximum electrical (E-) field and local SAR are determined by the feed conductor where the strongest current occurs; and (3) the coil is sensitive to loading, partly because of the small capacitors on the mode conductor. Therefore, we might be able to reduce the maximum local SAR (i.e., improve the B_1_^+^ SAR efficiency) and improve the coil robustness by elevating only the feed conductor and mode conductor. Note that the arm conductors would NOT be elevated to maintain the transmit efficiency and coil sensitivity. Therefore, unlike the conventional self-decoupled coil where all conductors are on the same planar surface [33], the proposed method here is a three-dimensional design in which the conductors are arranged intentionally on an uneven surface.

In this work, we first numerically investigated how the B_1_ efficiency and local SAR change with a spaced mode conductor and a spaced feed conductor but unchanged arm conductors on a water phantom. Then, we simulated the self-decoupled coil array with optimal lift-off distances on the human body and evaluated its performance. Next, a pair of transmit/receive self-decoupled coils with optimal lift-off distances was built and tested on the workbench. Finally, their B_1_^+^ efficiency and SNR, which are expected to be the same as those of the original self-decoupled coils, were tested and compared through MRI experiments.

## 2. Concept

Based on Ampere’s Law, the magnetic field generated by a straight conductor wraps around it. Therefore, magnetic fields from the feed conductor (orange in Figure 1A) and mode conductor (yellow in Figure 1A) are mainly along the z-direction, which contributes much less to B_1_. Meanwhile, the feed conductor with the strongest current generates the strongest E-field and thus determines the maximum local SAR. Figure 1B plots the magnitudes of the B_1_^+^ field (central axial slice) and E-field (coronal slice close to the coil) generated by these four individual conductors. Each conductor was driven with a series of current sources, with current magnitudes set to match those in a same-sized self-decoupled coil (10 × 10 cm^2^). The simulated B_1_ and E-fields clearly validated the assumption that B_1_ is unlikely to decrease when elevating feed and mode conductors, providing the foundation for this work. The concept simulations and the subsequent simulations for optimal lift-off distances were performed with an FEM-based Maxwell solver (HFSS, Ansys, Canonsburg, PA, USA) and an RF circuit simulator (Designer, Ansys, Canonsburg, PA, USA).

## 3. Methods

### 3.1. Simulation

We first numerically investigated how the E-field, B_1_, local SAR, B_1_^+^ SAR efficiency, coil impedance, and resonant frequency change when elevating the mode conductor. As shown in Figure 2A, pairs of 10 × 10 cm^2^ self-decoupled coils were modeled (conductor width 5 mm, coils are 5 mm apart) in Ansys HFSS. Similar to the design in the original self-decoupled coil [33], each coil has a parallel capacitor for matching (C_m_), two lumped components on the arm conductors for tuning (X_arm_), and five C_modes_ for decoupling. Various lift-off spacings of the mode conductor (D_mode_ in Figure 2A, from 0 cm to 4 cm in steps of 0.5 cm) were investigated, with all other conductors unchanged. In this assessment, a cuboidal phantom (30 × 15 × 15 cm^3^) was placed 1 cm below the coil as the loading. The electromagnetic (EM) properties of the phantom were chosen to be similar to those of human tissue and the same as those of a practical saline phantom, with conductivity σ = 0.6 S/m and relative permittivity ε_r_ = 78. The B_1_ efficiencies correspond to the B_1_ magnitudes normalized to the 1-watt input power. Considering that RF safety at ultrahigh fields is most likely limited by the local SAR instead of the global SAR [35,36,37], we did not investigate the global SAR changes. For each D_mode_, self-decoupled coils were first well-tuned/matched/decoupled when the coil-to-phantom distance was 1 cm. Then, the coils were moved closer or further away from the phantom, with no retuning or rematching. The resonance frequency shift and impedance matching were recorded when moving the coils.

Similarly, we numerically investigated how the E-field, B_1_^+^, B_1_^−^, and B_1_^+^ SAR efficiency change when elevating the feed conductor. Various lift-off spacings of the feed conductor (D_feed_ in Figure 2B, from 0 cm to 4 cm in steps of 0.5 cm) were investigated, with all other conductors unchanged. To ascertain whether elevating the feed conductor affects the decoupling performance, we also recorded the transmission coefficient (S_21_) between the elements of the self-decoupled coil array. To match the real case, these self-decoupled coils were all well-tuned, matched, and decoupled following the method described in our previous work [33].

Furthermore, we simulated a pair of self-decoupled coil arrays with an optimal D_feed_ of 2 cm on the human spine (Figure 2D) and compared them to the original self-decoupled coil array (Figure 2C). All coils are simulated for 7T, with an RF/Larmor frequency of 298 MHz. Both the local SAR and B_1_^+^ SAR efficiency were evaluated.

### 3.2. Coil Fabrication, Bench Test, and MRI Experiment

Based on the numerical investigations, we built a pair of self-decoupled coils with optimal D_mode_ and D_feed_. Details of the choices of D_mode_ and D_feed_ are provided in the Results section. For comparison, we also built a pair of original self-decoupled coils without any elevated conductors [33]. The values of all lumped elements were initially chosen based on the simulation results and then finely tuned by adjusting the trimmers (Johanson Manufacturing, 52 H Series, Boonton, NJ, USA) and air-core inductors (~25 nH). Bench tests were performed on an octagonal body phantom (~45 L, 1.24 g/L CuSO_4_ × 6H_2_O and 2.6 g/L NaCl) using a four-port Vector Network Analyzer (Keysight 5071C).

We measured B_1_^+^ maps on a body-shaped phantom (1 cm below coils) using the original and optimized self-decoupled coils. Individual B_1_^+^ maps were measured using the DREAM method [38] (field of view (FOV) = 400 × 224 mm^2^, TR = 1000 ms, voxel size = 2 × 2 mm^2^ and slice thickness = 10 mm) with the same input power. We also acquired low-flip-angle gradient echo (GRE, TR/TE = 1000/2.5 ms, FOV = 400 × 256 mm^2^, nominal flip angle = 15°, voxel size = 2 × 2 mm^2^ and slice thickness = 5 mm) images of individual coils for SNR assessment. SNR values were calculated from individual GRE images as SI/std(noise) × 0.655, where SI is the signal and std(noise) is the standard deviation of the noise maps. MRI experiments were performed on a Philips Achieva 7T whole-body scanner (Philips Healthcare, Best, The Netherlands).

## 4. Results

### 4.1. Simulation Results

Figure 3A shows the B_1_^+^, E-field, and local SAR maps with respect to D_mode_. B_1_^+^ maps and B_1_^+^ SAR efficiency were plotted on the central axial slice of the phantom, while E-field and local SAR were plotted on the coronal slice that was closest to the coil. We chose this slice to present the E-field and local SAR results because that is where the maximum E-field and SAR occur. Figure 3B,D plot the average B_1_^+^ and B_1_^+^ SAR efficiency at the surface and middle areas with respect to D_mode_. The average B_1_^+^ values were taken from two regions fixed in the phantom. The surface area (1.5 × 1.5 cm^2^) was immediately below the top surface of the phantom. The middle area (1.5 × 1.5 cm^2^) was 5 cm below the top surface of the phantom. Figure 4C plots the maxSAR_10g_ with different D_mode_. The B_1_ efficiency, B_1_^+^ SAR efficiency, and maxSAR_10g_ remain the same, even when D_mode_ increases to 4 cm. Figure 3E plots the largest frequency shifts when moving coils closer to or further away from the phantom. Coils exhibit less frequency shift as D_mode_ increases. This occurs because C_mode_ is not easily affected by the parasitic capacitance (between coil and loading) when the mode conductor and mode capacitors are elevated. Figure 3F,G plot the coil impedance (evaluated by S_11_ and S_21_) with respect to D_mode_. The impedance variation shows a similar trend to that of the frequency shift. However, the improvement in impedance is modest, which could be attributed to coils’ wide bandwidth (i.e., low quality factor), so the return loss does not change much when the resonant frequency is shifted. The curves in Figure 3E–G start to flatten when D_mode_ > 1 cm. In this work, we chose a D_mode_ of 2 cm for practical coil fabrication.

Figure 4A shows the simulated B_1_^+^, B_1_^−^, B_1_^+^ SAR efficiency, E-field, and local SAR maps with respect to D_feed_. Figure 4B plots the average B_1_^+^ at the surface and middle areas with respect to D_feed_. It is noted that B_1_^+^ efficiency was not affected when D_feed_ was 2 cm or less, with the B_1_^+^ variation <3%. This is also true for B_1_^−^ efficiency, as shown in Figure 4C. These results indicate that elevating the feed conductor would not impair the B_1_^+^ efficiency or receive SNR, as expected from the Concept section. Figure 4D shows the maxSAR_10g_ with different D_feed_s. Up to a 26% reduction of a maximum of 10 g SAR was observed when the feed conductor was elevated by 2 cm. The B_1_^+^ SAR efficiency (both the surface and middle areas) achieves the highest value when D_feed_ is approximately 2 cm, as shown in Figure 4E. A D_feed_ of 2 cm was thereby chosen for simulation on human spine and practical coil fabrication.

Figure 5 plots the B_1_^+^ efficiency, SAR_10g_, and B_1_^+^ SAR efficiency maps of the original self-decoupled coil [33] and optimized self-decoupled coil with a D_feed_ of 2 cm. B_1_^+^ maps and B_1_^+^ SAR efficiency are shown in the central axial slice, while SAR_10g_ is shown in the axial slice that is close to the feed port. We chose this slice to show SAR_10g_, as the maximum SAR_10g_ is located near the feed conductor. Compared with the original self-decoupled coil, the B_1_^+^ SAR efficiency of the optimized self-decoupled coil has 11.5% and 18.8% improvements at the surface and in the middle areas of the human body, respectively.

### 4.2. Bench Test and MRI Results

Figure 6A shows the fabricated original and optimized self-decoupled coils, and Figure 6B plots the measured scattering (S-) parameters when they were placed 1 cm above the phantom. Note that a cable trap was employed for each coil to suppress the common-mode current, but it is not shown in Figure 6A. Both the original and optimized coils achieve excellent decoupling performance, with S_21_ < −20 dB. Figure 6C,D compare their measured B_1_^+^ and SNR. As expected, coils without and with elevated conductors exhibit almost the same B_1_^+^ and SNR. Figure 6E shows the resonant frequency shift and coils’ input impedance with respect to the coil-to-phantom distance. Consistent with the simulation, the coil with elevated conductors demonstrated more robust tuning/matching performance, with the worst S_11_ of −11.3 dB (vs. −7.5 dB) and the largest frequency shift of 17.2 MHz (vs. 28.2 MHz).

## 5. Discussion

For all scenarios with different D_mode_s and D_feed_s, the coil isolation is at the same level of approximately −20 dB, as shown in Figure 7. This means only ~1% power crosstalk between the coils, which is sufficient for both Rx and Tx applications. This also indicates that the lift-off of the feed and/or mode conductor does not affect the decoupling performance and does not need to be considered during the optimization of D_mode_ and D_feed_.

The lift-off conductor design is mainly for transmit/receive applications where the self-decoupled coil is positioned close to the loading/tissue to maximize the receive sensitivity. For the Tx-only self-decoupled coil, which is typically several centimeters away from loading, there is significantly less improvement or even a decrease in B_1_^+^ SAR efficiency. Figure 8 shows how the B_1_ and maximum local SAR change when elevating the feed conductor for a self-decoupled coil that was already positioned 4 cm away from the loading. We noted that the B_1_^+^ SAR efficiency in the middle area increased by only ~1%, and this efficiency at the surface area even decreased for any lift-off distance.

It should be noted that simply elevating all conductors would significantly reduce the B_1_^+^ and B_1_^−^ efficiency and is therefore not recommended, as shown in the first row of Figure 9. It is interesting that as the lift-off distance increases, the B_1_^+^ (also B_1_^−^) efficiency at the surface area decreases much faster than that at the middle area. As a result, the B_1_^+^ SAR decreased by up to 23% in the surface area, while it slightly increased in the middle area when the whole coil was elevated by 4 cm.

For simplicity and clarity, we optimized D_mode_ and D_feed_ separately, which is reasonable considering that the SAR efficiency and coil robustness are separately determined by D_feed_ and D_mode_. In addition to the loop-type self-decoupled coil studied here, this elevated conductor design could be extended to loopole-mode [39,40] self-decoupled coils where the feed conductor orientates along the z-direction instead of perpendicular to the z-direction. In this case, the feed conductor plays two roles in B_1_^+^ SAR efficiency: its lift-off will change B_1_^+^ as well as the maximum SAR. Another parameter one can optimize for improved B_1_^+^ SAR efficiency and coil robustness is the dielectric constant of the substrate underneath the feed and mode conductors.

## 6. Conclusions

We propose a novel geometry to reduce the local SAR and improve the robustness of self-decoupled coils. A significant reduction in the maximum local SAR and a moderate improvement in the coil robustness were obtained by elevating the feed and mode conductors. We also confirmed that elevating these conductors does not impair the SNR or transmit efficiency.

## Figures and Tables

**Figure 1 sensors-23-01800-f001:**
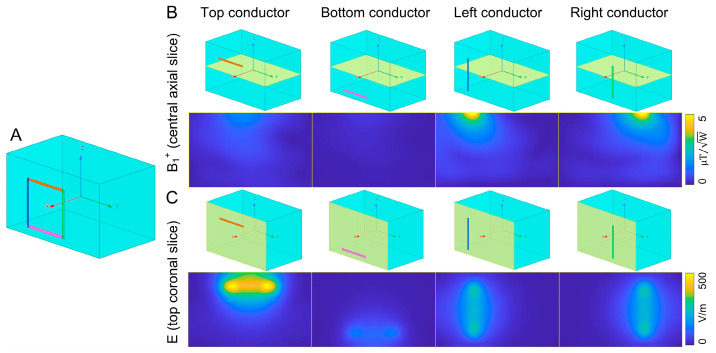
Simulation model (**A**) and results (**B**,**C**) demonstrate that the B_1_ and E fields are determined by different conductors in a self-decoupled coil. B_1_ is determined by the arm conductors (left and right conductors, blue and green in Figure 1A), while the maximum local E field is determined by the feed conductor (orange in Figure 1A).

**Figure 2 sensors-23-01800-f002:**
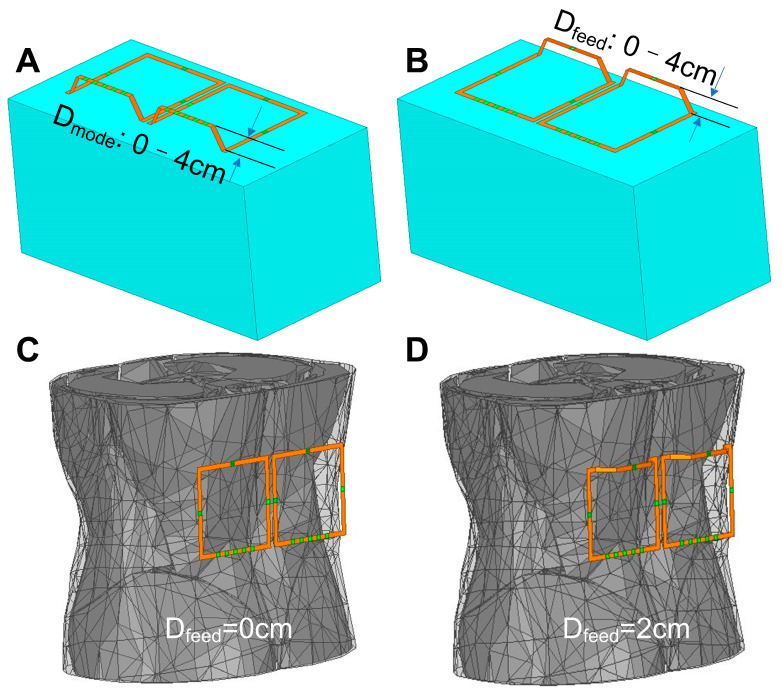
EM simulation models for optimizing lift-off distances of the mode conductor D_mode_ (**A**) and the feed conductor D_feed_ (**B**). Simulation models of the original (**C**) and optimized (**D**) self-decoupled coil on the human spine.

**Figure 3 sensors-23-01800-f003:**
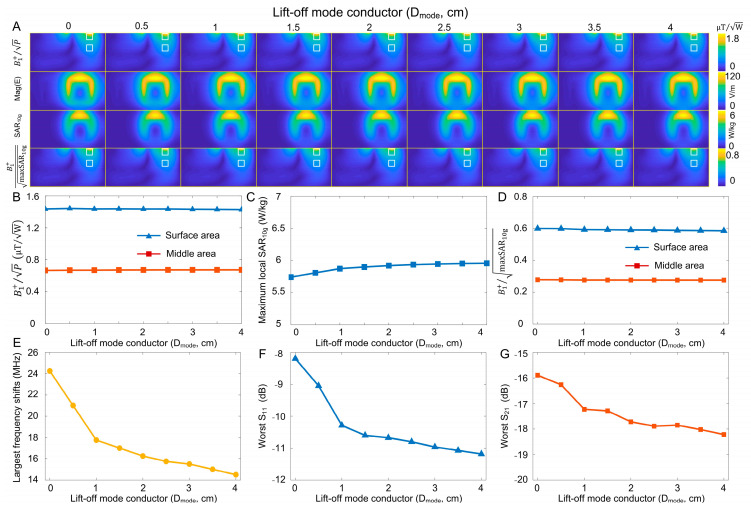
(**A**): Simulated B_1_^+^ efficiencies, E-fields, local SARs, and B_1_^+^ SAR efficiencies of self-decoupled coils with different lift-off distances of the mode conductor (D_mode_). (**B**–**D**): Plots of average B_1_^+^ efficiencies (**B**), maxSAR_10g_ (**C**), and average B_1_^+^ SAR efficiencies (**D**) versus D_mode_ at the surface and middle areas. (**E**): Largest resonant frequency shift (compared to 298 MHz) of self-decoupled coils with different D_mode_. (**F**,**G**): Worst S_11_ (**F**) and worst S_21_ (**G**) of self-decoupled coils when moving the coil closer or further away from the phantom.

**Figure 4 sensors-23-01800-f004:**
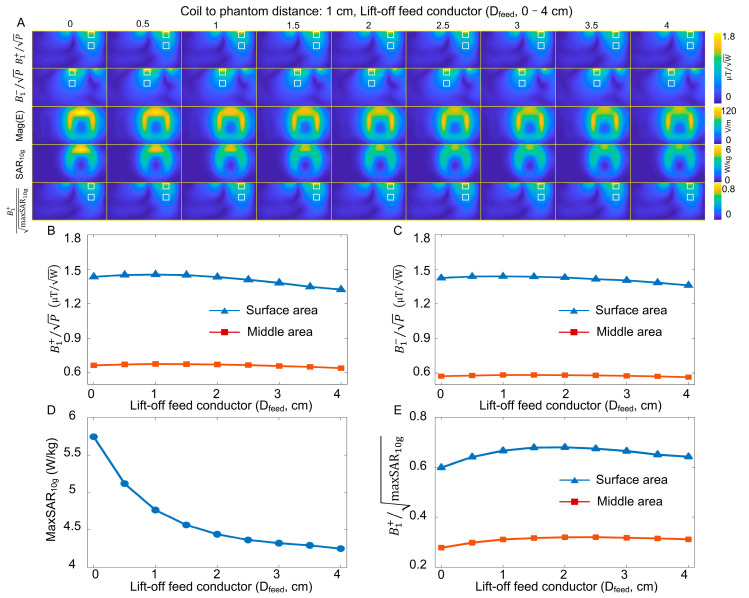
(**A**) Simulated B_1_^+^, B_1_^−^, E-fields, local SARs, and B_1_^+^ SAR efficiencies of self-decoupled coils with different lift-off distances of the feed conductor (D_feed_). (**B**,**C**) Plots of average B_1_^+^ and B_1_^−^ efficiencies versus D_feed_ at the surface and middle areas, respectively. (**D**) Plots of maxSAR_10g_ versus D_feed_. (**E**) Plots of average B_1_^+^ SAR efficiencies at the surface and middle areas.

**Figure 5 sensors-23-01800-f005:**
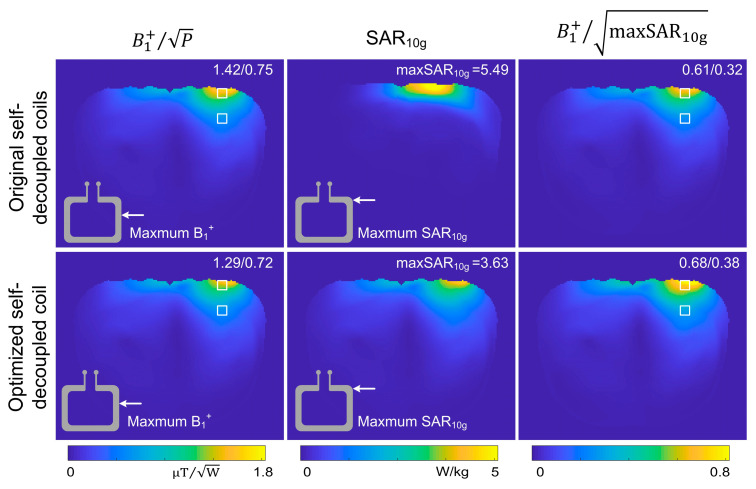
Comparison of simulated B_1_^+^ efficiencies, SAR_10g_, and B_1_^+^ SAR efficiencies between original and optimized self-decoupled coils on a human body model.

**Figure 6 sensors-23-01800-f006:**
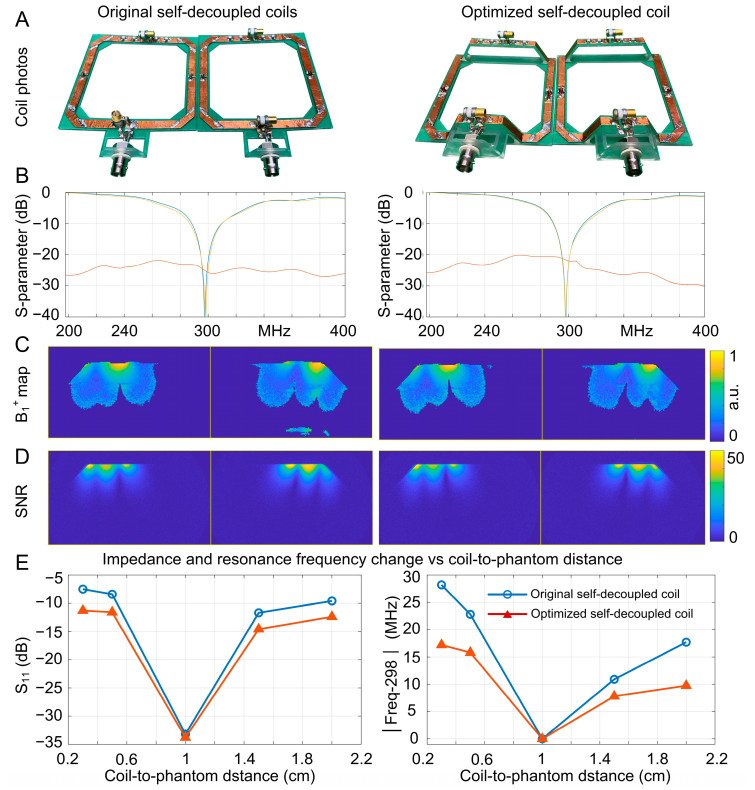
(**A**): Photographs of a pair of original (left) and optimized (right) self-decoupled coils. (**B**): Measured S-parameter plots versus frequency for the two pairs of coils. (**C**): Measured B_1_^+^ maps on the central transverse slice. (**D**): Measured SNR maps on the central transverse slice. (**E**): Measured coil impedance and resonance frequency shift versus different coil-to-phantom distances. Coils were first tuned and matched with a 1 cm separation from the phantom and then moved closer or further away from the phantom with no retuning or rematching.

**Figure 7 sensors-23-01800-f007:**
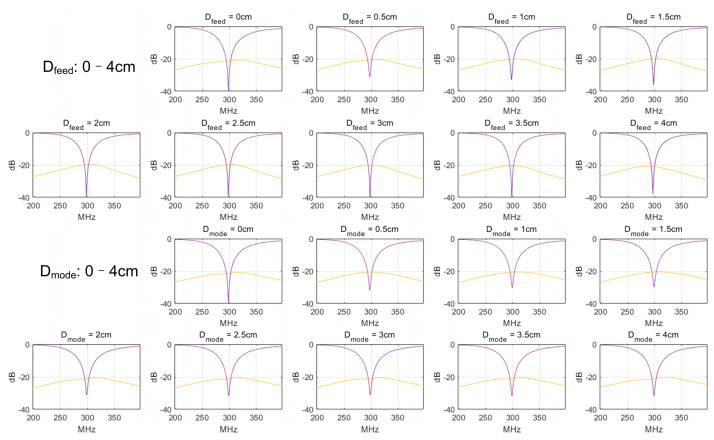
Simulated S-parameter plots of all pairs of self-decoupled coils with different D_feed_s and D_mode_s. For all scenarios, coils are well-tuned, matched, and decoupled, with S_11_/S_22_ < −30 dB and S_21_ < −20 dB.

**Figure 8 sensors-23-01800-f008:**
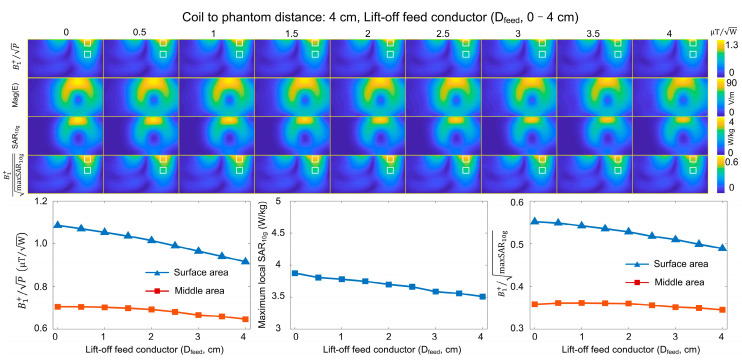
Simulated B_1_^+^ efficiencies, E-fields, local SARs, and B_1_^+^ SAR efficiencies of self-decoupled coils with different lift-off distances of the feed conductor (D_feed_) and coil-to-phantom distance equal to 4 cm.

**Figure 9 sensors-23-01800-f009:**
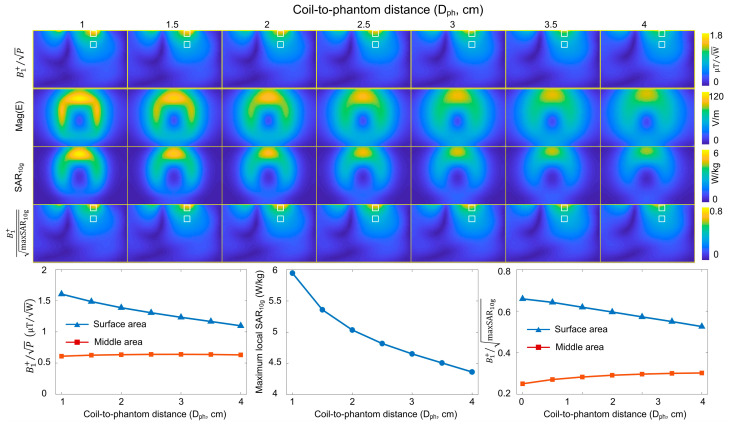
Simulated B_1_^+^ efficiencies, E-fields, local SARs, and B_1_^+^ SAR efficiencies of self-decoupled coils when elevating the whole coil, i.e., all conductors, instead of only the mode and feed conductors. An obvious B_1_ decrease was observed and therefore this design is not recommended.

## Data Availability

Not applicable.

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
