# Peer review of "Improving Specific Absorption Rate Efficiency and Coil Robustness of Self-Decoupled Transmit/Receive Coils by Elevating Feed and Mode Conductors"

_sensors, 2023, doi:10.3390/s23041800_

Round 1
Reviewer 1 Report
In this paper, the authors discuss a novel geometry to reduce the peak local SAR and improve the robustness of the self-decoupled coil. In general, the work in this paper is comprehensive. However, some improvements should be conducted before acceptance.
1. The literature review in section 1 is not sufficient. The authors should comprehensively review related work in detail.
2. The major difference between authors’ method and traditional method should be highlighted.
3. In the title and abstract, the word ‘Improving SAR efficiency’ is often discussed. However, the authors did not discuss the efficiency in their paper.
4. In Fig. 5, the original self-decoupled coil is used. The citations of original self-decoupled coil should be conducted.
Reviewer 2 Report
In this paper, the author proposed a novel geometry to reduce the peak local SAR and improve the robustness of the self-decoupled coil. The paper has sufficient theoretical basis, comprehensive simulation and reliable data, it is recommended to accept.
Reviewer 3 Report
There are some suggestions to improve this manuscript:
(1) Title: I suggest adding the full name of SAR (specific absorb ratio) in the title of this manuscript, to distinguish from the common-used SAR (synthetic aperture radar).
(2) Abstract: The full name of RF and SAR should be added in the abstract.
(3) Fig. 1: The arrows in the figure are blurry, it is recommended to increase the contrast.
(4) Line 137: The subscript of CuSO4 and H2O need to be corrected.
(5) Line 141, 143: the full name of FOV should be added.
(6) Figs. 3, 4, 5, 8: What is the meaning of the symbol “✔️” in the ylabels or the titles of subfigure?
